# DNA Methylation in Offspring Conceived after Assisted Reproductive Techniques: A Systematic Review and Meta-Analysis

**DOI:** 10.3390/jcm11175056

**Published:** 2022-08-28

**Authors:** Rossella Cannarella, Andrea Crafa, Laura M. Mongioì, Loredana Leggio, Nunzio Iraci, Sandro La Vignera, Rosita A. Condorelli, Aldo E. Calogero

**Affiliations:** 1Department of Clinical and Experimental Medicine, University of Catania, 95123 Catania, Italy; 2Department of Biomedical and Biotechnological Sciences (BIOMETEC), University of Catania, Torre Biologica, 95125 Catania, Italy

**Keywords:** DNA methylation, assisted reproductive technique, ART, offspring, epigenetics

## Abstract

**Background**: In the last 40 years, assisted reproductive techniques (ARTs) have emerged as potentially resolving procedures for couple infertility. This study aims to evaluate whether ART is associated with epigenetic dysregulation in the offspring. **Methods**. To accomplish this, we collected all available data on methylation patterns in offspring conceived after ART and in spontaneously conceived (SC) offspring. **Results.** We extracted 949 records. Of these, 50 were considered eligible; 12 were included in the quantitative synthesis. Methylation levels of *H19* CCCTC-binding factor 3 (CTCF3) were significantly lower in the ART group compared to controls (SMD −0.81 (−1.53; −0.09), I^2^ = 89%, *p* = 0.03). In contrast, *H19* CCCTC-binding factor 6 (CTCF6), *Potassium Voltage-Gated Channel Subfamily Q Member 1* (*KCNQ1OT1*), *Paternally-expressed gene 3* (*PEG3*), and *Small Nuclear Ribonucleoprotein Polypeptide N* (*SNRPN*) were not differently methylated in ART vs. SC offspring. **Conclusion**: The methylation pattern of the offspring conceived after ART may be different compared to spontaneous conception. Due to the lack of studies and the heterogeneity of the data, further prospective and well-sized population studies are needed to evaluate the impact of ART on the epigenome of the offspring.

## 1. Introduction

Couple infertility represents a relevant public problem, burdening psychological health, economic, and social aspects of couples looking for children. The last report of the World Health Organization (WHO) on 277 health surveys concluded that 48 million couples suffered from infertility in 2010 [1]. Nowadays, the global prevalence of infertility is, very likely, even higher. 

For the past 40 years, assisted reproductive techniques (ARTs) have emerged as potentially resolving procedures for couple infertility. They mainly include ovarian stimulation, fertilization (which can be achieved by in vitro fertilization (IVF) or by intracytoplasmic sperm injection (ICSI)), embryo culture, and embryo transfer. The first IVF baby was Louise Joy Brown who was born on 25 July 1978 [2]. Since then, ART has been broadly suggested to couples, even without being preceded by the attempt to identify and treat the etiological factors responsible for couple infertility [3]. The use of ICSI has increased from 36.4% in 1996 to 76.2% in 2012; although, the number of male-infertility cases did not change over time [4]. Moreover, some data indicate no real benefit from the use of ICSI (instead of IVF) in couples without male infertility, as the live birth rate seems 10% lower with ICSI than with IVF [5]. This may appear as an unjustified (or even blinded) use of ICSI [3].

In recent times, some data questioned the safety of ARTs. A retrospective longitudinal cohort study carried out on 797,657 children born in 2008–2019 reported a 1.23 times higher risk of hospitalization for any reason, 1.25 times higher risk of hospitalization for infection, and 1.25 times higher risk of hospitalization for allergy, in children conceived after ART compared to the spontaneously conceived (SC) siblings. These findings were not confirmed when a cohort of discordant siblings was used as a control [6]. Evidence from systematic reviews and meta-analyses suggested a trend towards a significantly increased risk of asthma (RR 1.31 (1.03–1.65)), but not allergies [7], a higher risk of autism [8] and of urogenital tract malformations (OR 1.42, (0.99–2.04)) [9] in offspring conceived after ART compared to controls. On the other hand, two recent longitudinal studies with a limited sample size failed in finding any difference in cardiometabolic profile and thyroid function between the ART and the non-ART cohort [10,11]. 

It has been speculated that the higher risk for adverse outcomes in the offspring conceived after ART could be due to epigenetic dysregulation [12,13]. In fact, the timing of ART procedures (ovarian stimulation, IVF/ICSI, embryo culture, and embryo transfer) coincides with crucial steps of embryo DNA methylation. DNA methylation takes place in the CpG islets, which are regions of the genome characterized by a large number of CpG dinucleotide repeats, and localized within the gene promoters. These regions are usually unmethylated and in specific circumstances (e.g., X-inactivation, genomic imprinting) undergo methylation to regulate gene expression. Indeed, hypermethylation generally interferes with chromatin accessibility, leading to gene silencing. In humans, more than 100 imprinted genes have been identified. They are clustered in differently methylated regions (DMR), which allow monoallelic gene expression [14]. During preimplantation development (day 1st to 5th), the embryo undergoes genome-wide demethylation and subsequent de novo methylation. The pattern of methylation of imprinted genes is not altered by this wave of reprogramming, thus ensuring their parent-specific expression [15]. 

An active debate is currently underway regarding the impact of ART on epigenetic reprogramming and imprinting in gametes and early embryos. In particular, there is no consensus on the possible effect of endogenous (gametes and embryo quality) and exogenous (e.g., light, cryopreservation, oxygen concentration, pH, temperature, culture media, mineral oil, humidity, centrifugation, etc.) factors in the ART setting responsible for increased reactive oxygen species (ROS) generation, which can lead to embryo epigenetic damage [16]. Furthermore, an abnormal methylation pattern has been reported in sperm from infertile men [17]. In turn, an altered methylation of imprinted genes at the sperm levels correlates with a poor ART outcome [18]. Whether the epigenetic risk of the ART-conceived offspring is due to the ART manipulation or to the epigenetic dysregulation of the gametes is still unknown. 

To assess whether ART is associated with an epigenetic dysregulation in the offspring, we performed a systematic review and meta-analysis, and gathered all the available data on methylation patterns in the offspring conceived after ART and in SC offspring. In line with a recently published systematic review and meta-analysis [19], data were grouped based on the examined tissue (placenta, cord blood, buccal smear, and peripheral blood).

## 2. Methods

The articles were selected through extensive searches in the PubMed and Scopus databases from their establishment until May 2022. The search strategy included the combination of the following Medical Subjects Headings (MeSH) terms and keywords: “assisted reproductive techn*”, “intracytoplasmic sperm injection”, “ICSI”, “in vitro fertilization”, and “epigenetic”.

The following search string was used to search the Scopus database: TITLE-ABS-KEY ((assisted AND reproductive AND techn*) OR (in AND vitro AND fertilization) OR (icsi) OR (intracytoplasmic AND sperm AND injection)) AND TITLE-ABS-KEY (epigenetic) AND (LIMIT-TO (DOCTYPE, “ar”)) AND (EXCLUDE (EXACT KEY WORD, “Animals”)) AND (EXCLUDE (SUBJAREA, “VETE”)) AND (EXCLUDE (LANGUAGE, “French”) OR EXCLUDE (LANGUAGE, “Russian”) OR EXCLUDE (LANGUAGE, “German”) OR EXCLUDE (LANGUAGE, “Chinese”)). The search was limited to human studies and only English articles were selected. The above-mentioned search strategy belongs to an unregistered protocol.

Studies were first evaluated for inclusion by reading their abstracts. When the abstract did not help to decide whether the study contained data relevant to our meta-analysis, the full text was read carefully. The identification of eligible studies was carried out independently by two different researchers (A.C. and R.C.). Any disagreements were resolved by a third author (A.E.C.). Others articles were manually extracted by searching the reference lists of the articles selected by the above keywords. 

The inclusion criteria are listed in Table 1. We considered for inclusion all studies that evaluated DNA methylation of offspring conceived using ARTs. Case reports, comments, letters to the editor, systematic or narrative reviews, and those studies that did not allow for extracting the outcomes of interest were excluded from the analysis. Two investigators (A.C. and R.C.) independently assessed the full text of the studies selected for eligibility. In case of disagreement, a third author (R.A.C. or A.E.C) decided against inclusion or exclusion after discussion.

The quality assessment of the articles included in this systematic review and meta-analysis was performed using the “Cambridge Quality Checklists” [20]. In detail, three domains are designed to identify high-quality studies of correlates, risk factors, and causal risk factors. The checklist for correlates consists of five items. Each item can be given a score of 0 or 1 for a total score of 5. This checklist evaluates the appropriateness of the sample size and the quality of the outcome measurements. The checklist for risk factors consists of three items; the selection of one of the 3 excludes the other two, with a maximum score of 3 points. This checklist assigns high-quality scores only to those studies with appropriate time-ordered data. Finally, there is the checklist for causal risk factors that evaluates the type of study design, assigning the highest score to randomized clinical trials (RCTs) and the lowest score to cross-sectional studies without a control group. The maximum score is seven. To draw confident conclusions about correlates, the correlate score must be high. This means that the sample size must be large and the outcome assessment must be adequate and reproducible. To draw confident conclusions about risk factors, both the checklists for correlates and risk factor scores must be high. Thus, the studies that allow the most reliable conclusions to be drawn are prospective studies. To draw confident conclusions about causal risk factors, all three-checklist scores must be high. Thus, in the absence of randomized clinical trials, confident conclusions can be drawn from studies with adequately controlled samples. Subgroup analyzes were performed based on the tissue in which methylation values were analyzed. Statistical heterogeneity was assessed by Cochran-Q and I^2^ statistics. For I^2^ ≤ 50%, the variation in the studies was considered homogenous and the fixed effect model was adopted. The random-effect model was used for I^2^ > 50%, underlying significant heterogeneity between studies. All *p* values ≤ 0.05 were considered statistically significant. The analysis was performed using RevMan software v. 5.3 (Cochrane Collaboration, Oxford, UK). The standard mean difference (SMD) with the 95% confidential interval (CI) was calculated for each outcome.

## 3. Results

Using the above-mentioned search strategy, we extracted 949 records. After the exclusion of 114 duplicates, the remaining 835 articles were assessed for inclusion in the systematic review. Of these, 167 were judged not pertinent after reading their title and abstract, 600 were excluded because they were reviews (n = 388), systematic reviews and meta-analyses (n = 4), and animal studies (n = 208). The remaining 68 articles were carefully read. Based on the inclusion and exclusion criteria, 15 articles were excluded because of the inability to extract the data required, and 3 were excluded because used miscarriage embryos [21,22,23]. Finally, 50 articles met our inclusion criteria and, therefore, were included in this meta-analysis (Figure 1).

Information on the design of the studies, the type of population and sample analyzed, the methodology for assessing DNA methylation, and the outcomes analyzed are summarized in Table 2. Analysis of study quality showed that all studies had a low to medium risk of bias (Table 3).

### 3.1. Qualitative Synthesis

All the results and limits of the studies included are summarized in Appendix A.

#### 3.1.1. Global Methylation

Since methylation at the level of transposable elements (TEs) occurs in around 50% of the human genome with a regulatory function for nearby genes, these can be used as an indirect marker of global methylation status [74]. With this premise, in the analysis of studies evaluating the impact of ART on global methylation of the DNA of the offspring, we included both studies assessing global DNA methylation and studies assessing methylation at the levels of TEs. Concerning this outcome, the studies showed considerable discordance. Indeed, in seven studies, variations were observed in the ART group compared to the group of SC offspring [25,26,33,38,42,47,51]. In detail, the studies generally showed the presence of hypomethylation in both global DNA and at the level of TEs in the group conceived by ART compared with that in the group of SC offspring [25,26,33,47,51]. In one study, hypermethylation at the level of cord blood and hypomethylation at the level of the placenta was observed in the ART group compared to the SC group [42]. In another study, hypermethylation was observed in the LUMA assay and hypomethylation in the *LINE1* assessment in the ART group compared to the SC group [38]. However, in other eight studies, no difference was observed between global methylation rates in the ART and control groups [27,31,32,35,46,49,54,56].

#### 3.1.2. Methylation of Imprinted Genes

With regard to the involvement of imprinted genes, 10 studies showed no alteration in the imprinted genes analyzed [24,43,45,53,55,58,61,65,66,71], while another 11 studies showed alterations in at least one of the imprinted genes [25,26,30,39,41,47,52,59,67,68,70]. In particular, among the main genes evaluated in the various studies, we encounter *H19*, *Insulin-like growth factor 2* (*IGF2*), *Small Nuclear Ribonucleoprotein Polypeptide N* (*SNRPN*), *Mesoderm Specific Transcript* (*MEST*), the *Potassium Voltage Differentially Methylated Region 1* (*KvDMR1*) region of the *Potassium Voltage-Gated Channel Subfamily Q Member 1 Opposite Strand*/*Antisense Transcript 1* (*KCNQ1OT1*) gene, and *Maternally Expressed Gene* (*MEG3*). For the *H19* gene, five studies showed hypomethylation in the differentially methylated regions (DMRs) of this gene in the ART group compared to the SC group [25,26,47,52,59]. Instead, one study showed hypermethylation [30], and another do not specify the type of aberration [67]. In contrast, eight studies observed no difference [43,55,58,61,65,66,71]. As for the DMRs of its complementary gene, *IGF2*, four studies showed no difference in methylation between the ART and the SC control group [24,55,58,70]. As for the *MEST* gene, two studies showed no difference in methylation levels between the ART group and SC controls [24,66], while two studies found it was hypomethylated in the ART group than SC group [52,59]. For the *SNRPN* gene, three studies showed no difference in its methylation in the ART-conceived offspring compared to SC controls [55,65,66], while two studies found hypermethylation in the ART group compared to SC controls [59,70]. Regarding methylation of KvDMR1 or other regions of the *KCNQ1OT1* gene, three studies found an abnormal methylation of this gene in the ART vs. the spontaneously-conceived offspring [30,39,59]. In detail, two studies found it hypomethylated in the ART group compared with the SC group [30,59], while 1 study found it hypermethylated [39]. On the contrary, five studies did not find any difference between the two groups [43,55,58,65,66]. Similar heterogeneity in results was also observed for other genes, such as MEG3 [41,52]. 

#### 3.1.3. Role of ART Protocol and Technique

Since numerous protocols of ART (controlled ovarian stimulation (COS), fresh vs. frozen embryo transfer (ET), in vitro fertilization (IVF) vs. intracytoplasmic sperm injection (ICSI), embryo transfer day, and culture medium used [16]) have been implicated in epigenetic changes, we analyzed the results of the studies evaluating the impact of the individual ART processes on DNA methylation. 

Regarding the studies that have evaluated the role of COS, four studies concluded it could play a predominant role in causing epigenetic changes [30,40,41,59], while three conclude that COS is not responsible for these alterations [32,48,72]. 

As for fresh vs. frozen ET, most of the studies that analyzed the difference in methylation between the two methods concluded that fresh ET correlates with major alterations compared to the frozen one [26,36,38], two studies concluded that there is no difference between the two methods [40,54] and one study instead found that cryopreservation could be associated with a greater carcinogenic risk [31]. All studies that analyzed the difference in global methylation of DNA or imprinted genes according to the day of ET found no association [38,40,69]. 

Only two studies evaluated the impact of the culture medium, with conflicting results [54,55]. 

Finally, as regards the difference between the various techniques used in ART, only five studies found that ICSI is associated with greater alterations than IVF [26,29,47,70] or intrauterine insemination (IUI) [31], while three studies concluded that IVF is associated with a greater DNA methylation aberration than ICSI [25,48,68]. However, in most of the studies, this difference was not evaluated and no difference was found between the two methods [54,62].

The results of the qualitative analysis are shown in Appendix A.

#### 3.1.4. Role of Parental Age

Because parental age can also influence gamete quality and thus promote the occurrence of epigenetic abnormalities that can then be transmitted via ART [75], we evaluated the number of studies that reported parental age and performed an adjusted analysis taking it into consideration. We found that only nine studies reported paternal age [32,33,38,42,44,49,53,57,64]. However, three of them did not perform an adjusted analysis by paternal age [33,42,44]. On the other hand, with regard to maternal age, 14 of the 50 included studies did not report the maternal age and, therefore, did not consider it in the adjusted analysis [25,34,36,48,50,54,55,60,62,63,65,69,72,73]. However, in four other studies, although reported, the analysis would not appear to be corrected by parental age [44,46,66,67].

#### 3.1.5. Role of the Etiology of Infertility

Among all the included studies, only 13 corrected the analysis by excluding the male factor or directly analyzed the role of infertility [31,32,35,37,45,50,55,56,63,64,65,69] with conflicting results in this case as well. In detail, Chen and colleagues showed that both ART methods and infertility per se could lead to alterations in DNA methylation [31]. Another study also showed that, by correcting the analysis taking into account the father’s sperm concentration, the ART group still had significant differences in methylation levels compared to the group of SC children [35]. Likewise, White and colleagues observed that two embryos generated by ICSI with donor sperm, therefore healthy, also had methylation aberrations [69]. Finally, Song and colleagues comparing a group of children born from ART by infertile fathers and children born from ART with fathers without infertility identified very similar methylation abnormalities between the two groups that, in turn, differed significantly from those of SC children [64]. These results seem to confirm the role of the methods per se in causing epigenetic alterations regardless of the presence of the underlying paternal infertility. However, other studies have come to the opposite conclusion. Choufani and colleagues showed that the methylation differences in the ICSI/IVF group were seen to be closely related to male infertility and paternal age [32]. In another study, Litzky and colleagues showed that only the group of children conceived by parents with underlying infertility (one or both parents) had methylation alterations, compared to the IVF and SC groups. Therefore, the alterations in methylation observed in children conceived by ART could also be partly attributed to underlying infertility and, therefore, to the alteration of the gametes used for the technique [45].

### 3.2. Quantitative Synthesis

A total of 12 studies [19,33,39,52,55,58,59,61,63,65,66,71] were included in the quantitative analysis. Methylation levels of the following genes could be meta-analyzed: *H19* CCCTC-binding factor 3 (CTCF3), *H19* CTCF6, *KCNQ1OT1*, Paternally Expressed Gene 3 (*PEG3*), and *SNRPN*. Moreover, also methylation levels of the Arthrobacter luteus (Alu), *Long Interspersed Nuclear Elements* (*LINE*) (most investigated TEs) could be meta-analyzed. 

*H19* CTCF3 methylation levels were significantly lower in the ART group compared to controls (SMD −0.81 (−1.53; −0.09), I^2^ = 89%, *p* = 0.03). Subgroup analysis showed a significantly lower methylation in placenta (−0.53 (−0.83, −0.22), I^2^ = 0%, *p* < 0.05) and buccal smear (1.61 (−3.09, −0.12), I^2^ = 92%, *p* = 0.03) (Figure 2). In contrast, *H19* CTCF6 methylation was not significantly different between ART and controls (0.02 (−0.23, 0.26), I^2^ = 66%, *p* = 0.89). Furthermore, the subgroup analysis showed no difference in the methylation levels of each tissue (Figure 3). Similarly, *KCNQ1OT1* (−0.15 (−0.38, 0.09), I^2^ = 71%, *p* = 0.22) (Figure 4), *PEG3* (−0.15 (−0.38, 0.09), I^2^ = 71%, *p* = 0.59) (Figure 5), *SNRPN* (−0.02 (−0.19, 0.15), I^2^ = 37%, *p* = 0.82) (Figure 6), were not differently methylated in ART vs. SC control offspring.

## 4. Discussion

The development of ART was a huge step forward in the treatment of couple infertility, leading to the birth of numerous newborns. Every year, more than 200,000 children are born through ART worldwide [76]. However, Barker’s theory of Developmental Origins of Health and Disease (DOHaD), according to which alterations in the microenvironment of conception can cause long-term damage, particularly cardiovascular and metabolic diseases, has raised concerns that the techniques used may alter the imprinting and, therefore, lead to long-term disorders [77].

DNA methylation reprogramming occurs in two different moments. The first reprogramming concerns the gametes. The genome of primordial germ cells is completely demethylated as they enter the genital crest, and then undergo sex-specific de novo methylation with the establishment of specific methylation patterns for imprinted genes. The second wave of genome-wide demethylation and subsequent de novo methylation occurs during preimplantation development. Only the methylation pattern of imprinted genes is not altered by this second wave of reprogramming, which ensures their parent-specific expression and activity throughout development [15]. The latter occurs when ART procedures are carried out (Figure 7). 

Several studies have shown a higher prevalence of disorders associated with altered imprinting, such as Beckwith–Wiedemann syndrome (BWS) and Silver–Russell Syndrome (SRS), in ART-born children [78,79]. In this context, some studies have evaluated the effects of ART on DNA methylation. In particular, ART could influence the methylation and, therefore, the expression of imprinted and non-imprinted genes that may be involved in insulin signaling pathways and adipocyte differentiation, suggesting a role of these procedures in the development of diabetes and future obesity [42].

Furthermore, ART can alter the expression of genes involved: (i) in the development of the nervous and immune systems [21]; (ii) in the susceptibility of cancer development [28]; and (iii) also in future fertility, such as *Spermatogenesis* and *Centriole Associated 1 Like* (*SPATC1L*) gene, which encodes for speriolin [36]. The altered methylation of some genes could also be associated with a worsening of short-term fetal outcomes (e.g., birth weight) and gestational complications. In this regard, it has been shown that ART may increase the risk of preeclampsia due to hypomethylation of the *Angiotensin II Receptor Type 1* (*AGTR1*) gene, which results in an upregulation of its levels. In turn, this altered methylation pattern could be due to reduced expression of the DNA methyltransferase 3a (*DNMT3a*) gene, which is responsible for de novo DNA methylation. All of this makes the umbilical veins more sensitive to the effects of angiotensin II, since AGTR1 is the main mediator of vasoconstriction [73]. In addition, ART may be associated with reduced methylation of the promoter of the *MEG3* gene. This leads to the higher expression of *endothelin 1* and *endothelial nitric oxide synthase* (*eNOS*), which increase vasoconstriction. This would explain the increased blood pressure that some studies have found in children born from ART [41]. Finally, the hypomethylation of the *KvDMR* gene, in turn, associated with an increase in *Cyclin Dependent Kinase Inhibitor 1C* (*CDKN1C*), impairs growth. Likewise, alterations in the methylation of H19/IGF2 DMRs or other genes such as *MEST*, can alter fetal growth and increase the prevalence of low birth weight in children born by ART [30]. 

This systematic review aims to analyze the evidence presented to date in the literature on the effects of ART procedures on the methylation of global DNA and specific imprinted genes. Our quantitative synthesis showed a significantly reduced methylation of *H19* CTCF3 in the offspring conceived after ART compared to SC. However, there was an inter-study heterogeneity, which could be partly explained by the different samples used for the analysis (placenta, cord blood, or peripheral blood), the different methods to evaluate DNA methylation, and the different sample sizes. For imprinted genes, another reason for heterogeneity is the difference in the region of the gene analyzed for methylation. Furthermore, as suggested by the study of Turan and colleagues, given the extreme variability not only inter- but also intra-individual in DNA methylation, a role in the heterogeneity of the results could also be given by the region in which the placenta biopsy was performed [67]. Finally, there is often a lack of standardization regarding the ART process used. About the latter point, very few studies have specifically examined the impact of the various steps of ART on DNA methylation.

The most investigated aspect is the COS. Several studies have attributed the DNA methylation abnormalities found to the high estrogen levels achieved during COS [59]. Indeed, Jiang and colleagues found that the expression levels of the *MEG3* and *endothelin 1* genes directly correlated to estrogen levels [41]. Similarly, incubation of human trophoblast 8 (*HTR8*) cells with high estrogen levels resulted in hypomethylation of the *KvDMR1* gene after 24 h of incubation and hypermethylation of H19 DMR after 48 h [30]. Other studies that have evaluated the impact of various ART methods, including those where no major manipulation of embryos and gametes was made (e.g., IUI and gamete intra-fallopian transfer (GIFT)), found no difference [40,54]. However, a difference in methylation profiles was found when comparing the ART group in general with that of SC infants [54]. Finally, comparative studies between fresh ET and frozen ET would also seem to confirm a prominent role of COS, since, in fresh ET the higher estrogen levels reached would cause a dysregulation of the endometrial microenvironment, which according to the DOHaD theory would then be responsible for the long-term damage on the embryo [26,36,38]. However, although there is a lot of evidence in favor of COS’s role, some studies have disproved this hypothesis. For example, Luo and colleagues compared a group of children conceived by IVF and ICSI with a group of children conceived only by COS, showing that only the former was associated with hypomethylation of *H19* and hypermethylation of *IGF2* DMR2 and *SNRPN* DMR [48]. Another study showed no effect of in vitro maturation (IVM) and COS on the methylation of specific imprinted genes [72]. Finally, another study comparing methylation alterations in a group of children conceived by IUI/COS and a group by IVF/ICSI showed that there was different DNA methylation in the IVF/ICSI group, suggesting that COS, common to both groups, is not the real culprit behind the observed differences. Therefore, the difference could relate to the greater manipulation of gametes and embryos with the more invasive techniques [32]. 

Another major bias present in most studies is the absence of correction of the analysis for the paternal factor of infertility. Accordingly, only 12 corrected the analysis by excluding the male factor or directly analyzed the role of infertility [31,32,35,37,45,50,55,56,63,64,69], with conflicting results in this case as well. Similarly, many articles did not even consider paternal age, which has instead been seen to correlate with offspring well-being through three basic mechanisms: genetic mutations, telomere length, and epigenetic changes in DNA, and protein expression [75].

Finally, another important limitation of the included studies is that almost all of them and the data analysis are cross-sectional. There are no data to predict whether the methylation changes found in newborns are associated with the development of abnormalities in these children in the long term. The only study with longitudinal data showed a higher prevalence of *SNRPN* DMR hypermethylation in children conceived by ICSI that does not change after 7 years of age, suggesting that these changes may be stable and perpetuate over time [70]. 

To the best of our knowledge, this is the second meta-analysis evaluating the methylation differences in offspring conceived after ART vs. SC. A recent systematic review and meta-analysis of 51 studies found no difference in *H19* methylation. In contrast, they found different methylation in the *Paternally Expressed Gene 1* (*PEG1*)/*MEST* region. However, the data were analyzed separately for each tissue, thus limiting the amount of data for each gene evaluated [19]. The evidence coming from our systematic review and meta-analysis suggests that *H19* CTCF3 methylation levels are significantly lower in the ART offspring compared to controls. In contrast, *H19* CTCF6, *KCNQ1OT1*, *PEG 3*, and *SNRPN* were not differently methylated in ART than vs. SC. 

## 5. Conclusions

Nowadays, ART is widely used for male and female infertility. Emerging evidence indicates a higher health risk in ART than in SC offspring. Despite this, the exact link between ART and the increased risk of epigenetic abnormalities predisposing to the development of diseases is unclear. The debate is still ongoing as some studies found a different global DNA methylation and the methylation of genes imprinted in ART-conceived offspring compared to controls. However, other studies have not confirmed this evidence, suggesting the absence of any epigenetic aberration. Using a defined search strategy, we extracted 949 records. Among them, 50 were considered eligible. We found that *H19* CTCF3 methylation levels were significantly lower in the ART group compared to controls, in the presence of significant inter-study heterogeneity (SMD −0.81 (−1.53; −0.09), I^2^ = 89%, *p* = 0.03). In contrast, *H19* CTCF6, *KCNQ1OT1*, *PEG3*, and *SNRPN* were not differently methylated in ART vs. SC offspring. The heterogeneity of the results could be due to the lack of correction of the data for parental (male or female) infertility, the limited sample size, the retrospective design of almost all studies, the different methods used to analyze the methylation rate (including the different DMRs studied) and, finally, also the different regions where the placenta biopsy was performed. Therefore, further prospective and well-sized population studies are needed to evaluate the impact of ART on the epigenome of the offspring. Furthermore, it is necessary to clarify the contribution of the different protocols and techniques used during ART to the etiology of epigenetic aberrations. Finally, the weight of the presence of maternal and/or paternal infertility in causing alterations in methylation deserves to be further explored.

## Figures and Tables

**Figure 1 jcm-11-05056-f001:**
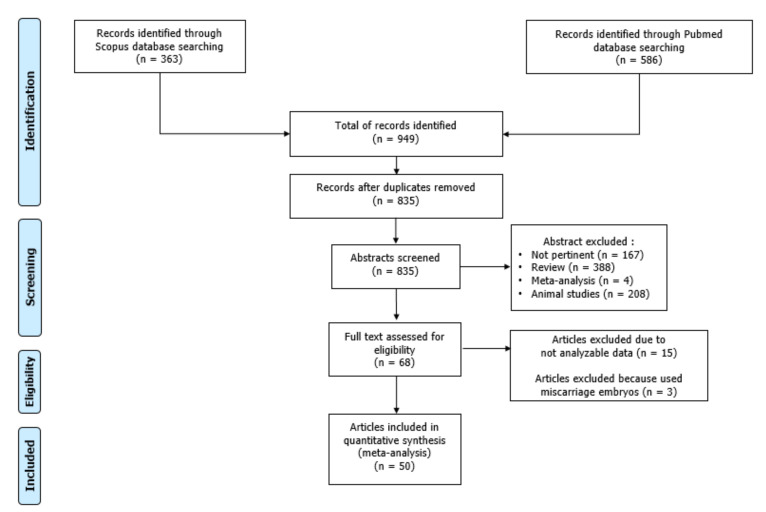
Flowchart of the included studies.

**Figure 2 jcm-11-05056-f002:**
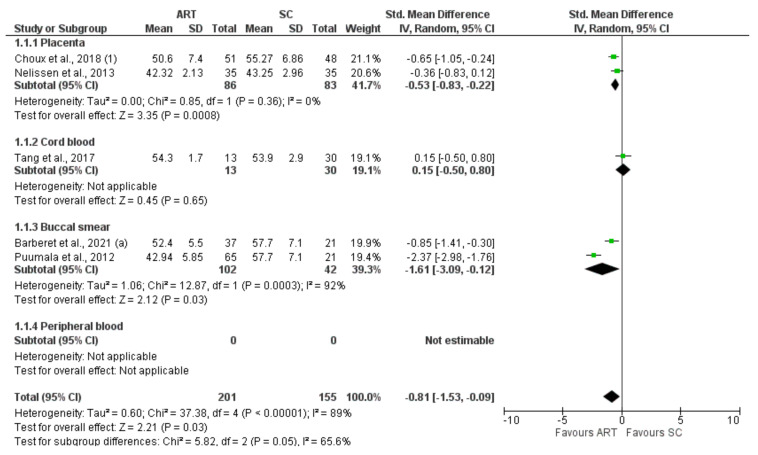
Methylation levels of *H19* CTCF3 [25,33,52,58,65]. ART, assisted reproductive technique; SC, spontaneous conception.

**Figure 3 jcm-11-05056-f003:**
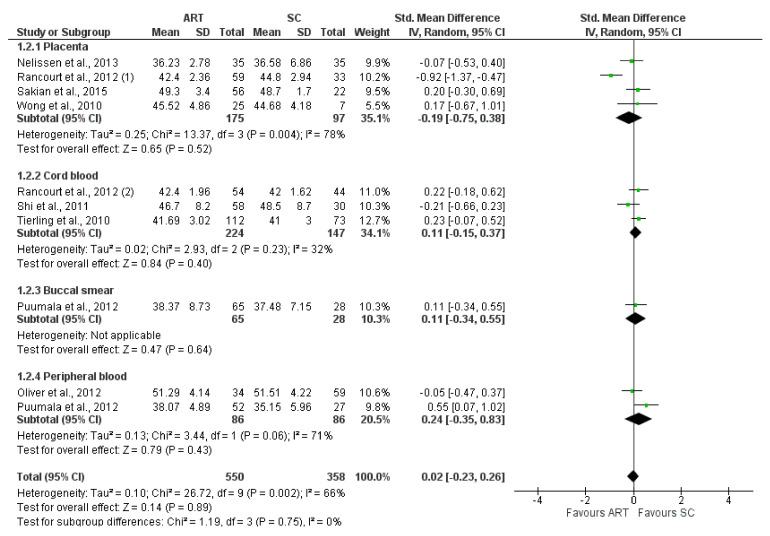
Methylation levels of *H19* CTCF6 [52,55,58,59,61,63,66,71]. ART, assisted reproductive technique; SC, spontaneous conception.

**Figure 4 jcm-11-05056-f004:**
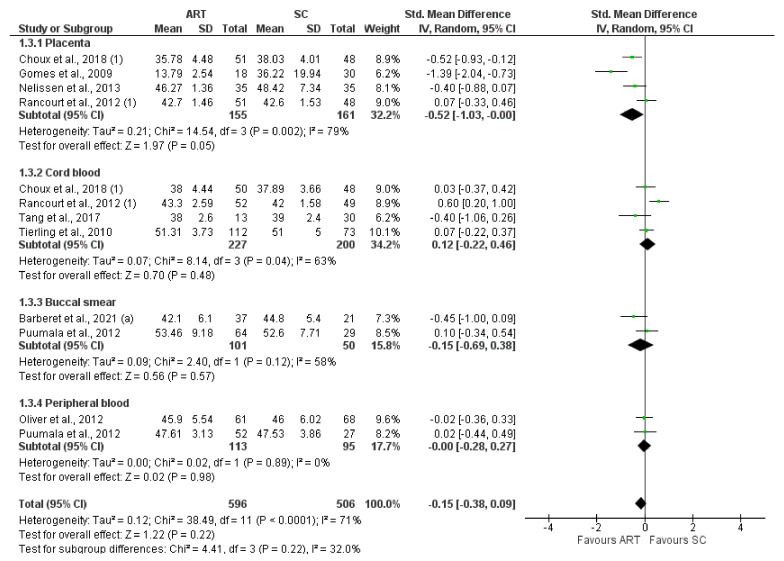
Methylation levels of *KCNQ1OT1* [25,33,39,52,55,58,59,65,66]. ART, assisted reproductive technique; SC, spontaneous conception.

**Figure 5 jcm-11-05056-f005:**
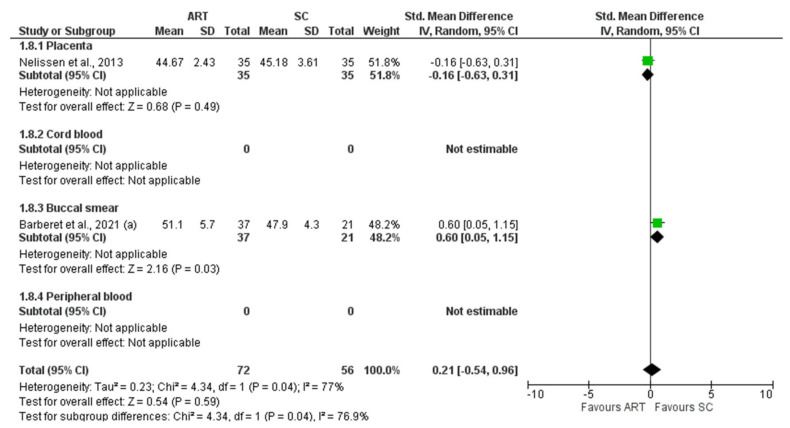
Methylation levels of *PEG3* [25,52]. ART, assisted reproductive technique; SC, spontaneous conception.

**Figure 6 jcm-11-05056-f006:**
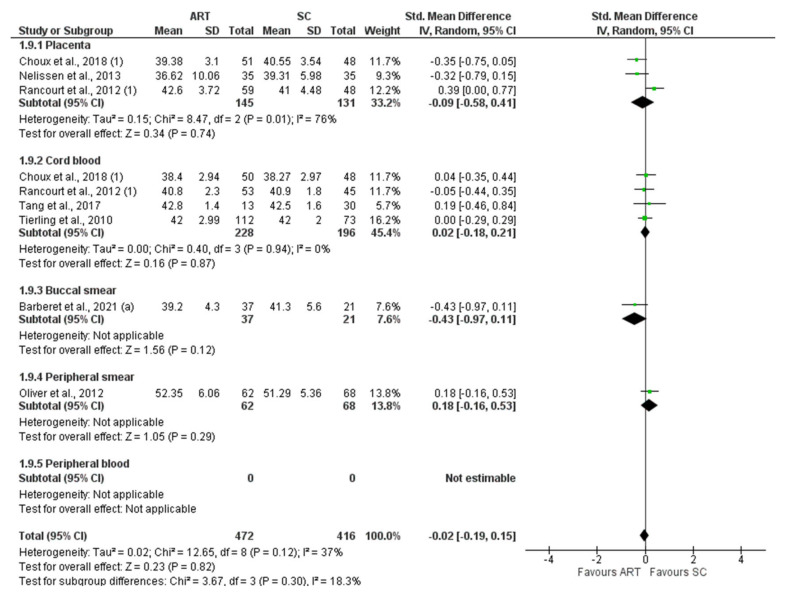
Methylation levels of *SNRPN* [25,33,52,55,59,65,66]. ART, assisted reproductive technique; SC, spontaneous conception.

**Figure 7 jcm-11-05056-f007:**
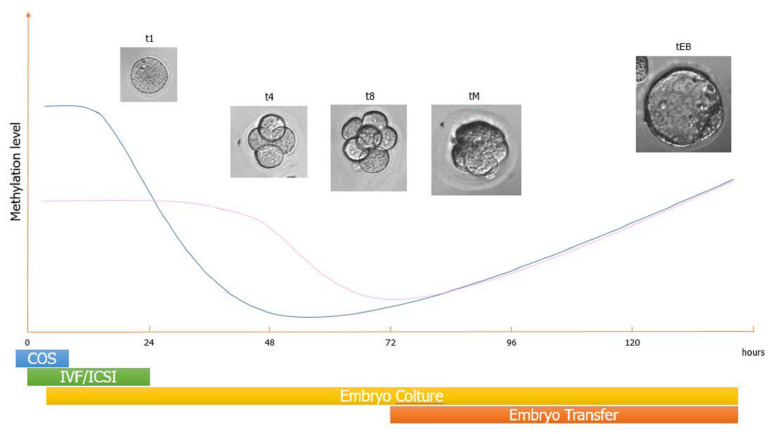
Timing of the methylation pattern of paternal and maternal alleles during human embryogenesis. After fertilization, the embryo undergoes the first wave of global demethylation, followed by de novo methylation. Only the imprinted genes escape epigenetic reprogramming. The timing of these events is concomitant with that of in vitro fertilization (IVF), intracytoplasmic sperm injection (ICSI), embryo culture, and embryo transfer. COS controlled ovarian stimulation.

**Table 1 jcm-11-05056-t001:** Inclusion criteria.

	Inclusion	Exclusion
Population	Human offspring	/
Intervention	ART (including IVF, ICSI, IUI, FET, ET, COS, OI)	/
Comparison	SC	/
Outcome	Methylation statuses of both imprinted and non-imprinted genes, global DNA methylation, evaluated in any kind of tissue and at any age	Aborted embryos
Study type	Observational, cohort, cross-sectional, and case-control	Case reports, comments, letters to the editor, systematic or narrative reviews, in vitro studies, studies on animals

**Abbreviations**. ART, assisted reproductive techniques; COS, controlled ovarian stimulation; ET, embryo transfer; FET, frozen embryo transfer; ICSI, intracytoplasmic sperm injection; IUI, intrauterine insemination; IVF, in vitro fertilization; OI, ovulation induction; SC, spontaneous conception.

**Table 2 jcm-11-05056-t002:** Main features of the included studies.

Author and Year	Study Design	Etiology of Infertility (M/F)	Paternal/ Maternal Age (y)	Tissue	Timing	ART Group	SC Group (Parents’ Fertility Status)	Outcome Assessed	Methylation Evaluation Method
Argyraki et al., 2021 [24]	Cross-sectional	NR	NR/35.2 ± 3.12	Cord blood	Birth	10	30 (10 delivered naturally, 10 by cesarean section in head position, 10 by cesarean section in breech position) (NS)	*IGF2*, *MEST*, *PEG10*	Methylation-specific PCR
Barberet et al., 2021 [25]	Cross-sectional	NR	NR	Buccal smear	Childhood	37 (16 IVF, 21 ICSI)	21 (fertile)	*H19*, *SNURF*, *PEG3 KCNQ1*, *LINE1*, *AluYa5*	Pyrosequencing and EPIC array
Barberet et al., 2021 [26]	Cross-sectional	NR	NR/ICSI-ET: 33.1 ± 3.9; ICSI-FET: 31.3 ± 5.1; SC: 29.1 ± 3.6	Placenta Cord blood	Pregnancy Birth	118 (66 IVF/ICSI-ET, 52 IVF/ICSI-FET)	84 (fertile)	*H19/IGF2*, *KCNQ1OT1*, *SNURF*, *LINE1*, *HERV-FRD*	Pyrosequencing
Camprubì et al., 2013 [27]	Cross-sectional	NR	NR/ART: 36.2 ± 5.0; SC: 33.3 ± 5.4	Placenta Cord blood	Birth	73	121 (NS)	*LINE1*, *AluYbU*, a-satellite repeats, and the promoters of *SLC2A3*, *PLA2G2A*, and *VEGFA*	Illumina Goldengate methylation array and pyrosequencing
Caramaschi et al., 2011 [28]	Cross-sectional	NR	NR/ART: 29.65 ± 4.41; SC: 28.84 ± 4.83	Placenta	Birth	205	2439 (NS)	Global DNA methylation	Illumina Methylation 450k BeadChip Array
Castillo-Fernandez et al., 2017 [29]	Cross-sectional	NR	NR/NR in total sample	Cord blood	Birth	47	60 (NS)	Global DNA methylation	MeDIP-sequencing
Chen et al., 2018 [30]	Cross-sectional	NR	NR/ART: 32.9 ± 3.3; SC: 31.5 ± 4.3	Placenta	Birth	35 (COS-FET)	37 (NS)	*CDKN1C*, *IGF2*	Bisulfite sequencing
Chen et al., 2020 [31]	Cross-sectional	NR	NR	Cord blood	Birth	NR	NR	Global DNA methylation	RRBs for DNA methylome and CHIP for histone modifications
Choufani et al., 2018 [32]	Cross-sectional	M in 12/40 F in 6/40	ART: 34.5 ± 4.3; SC: 33.0 ± 3.8/ART: 34.7 ± 7.0; SC: 36.0 ± 5.3	Placenta	Birth	23 (18 ICSI, 5 IVF), 11 IUI, 10 (more than one technique)	44 (fertile)	Global DNA methylation	Illumina Human Methylation 450 BeadChip array and pyrosequencing
Choux et al., 2018 [33]	Cross-sectional	NR	ART: 33.7 ± 5.7; SC: 31.9 ± 5.2/ART: 31.1 ± 5.3; SC: 29.4 ± 4.0	Placenta Cord blood	Birth	51	48 (fertile)	*ERVFRD1*, *ERVW1*, *LINE1*, *AluYa5*	Bisulfite pyrosequencing
DeBaun et al., 2003 [34]	Observational uncontrolled	NR	NR	Peripheral blood	Children	6 (ICSI)	/	*LIT1*, *H19*	Southern blot
El Hajj et al., 2017 [35]	Cross-sectional	NR	NR/IVF: 34.3 ± 4.5; ICSI: 34.0 ± 3.9; SC: 30.2 ± 5.9	Cord blood	Birth	48	46 (NS)	Global DNA methylation	Illumina 450 k Methylation Array and pyrosequencing
Estill et al., 2016 [36]	Cross-sectional	NR	NR	Peripheral blood	Children	76 (38 ICSI-ET, 38 ICSI-FET), 18 IUI	43 (NS)	Global DNA Methylation	Illumina Infinium Human Methylation 450 BeadChip Array
Feng et al., 2011 [37]	Cross-sectional	NR	NR/IVF: 31.0 ± 3.7; ICSI: 29.1± 3.6; SC: 29.7 ± 4.2	Cord blood	Birth	60 (30 IVF, 30 ICSI)	60 (NS)	*L3MBTL*	Bisulfite sequencing
Ghosh et al., 2017 [38]	Cross-sectional	NR	ART: 36.9 ± 5.7; SC: 33.3 ± 5.2/ART: 34.7 ± 3.6; SC: 32.2 ± 4.8	Placenta	Birth	182	77 (NS)	*LINE 1*	Pyrosequencing for LINE1 and
Gomes et al., 2009 [39]	Cross-sectional	M = 7 F = 7 M + F = 4	NR/ART: 32.3 ± 4.27	CSV, cord blood, placenta, peripheral blood	Birth, children	12, 6	8, 22, 3 (NS)	*KvDMR1*	Methylation-specific PCR
Ji et al., 2018 [40]	Cross-sectional	NR	NR/Total: 30.1 ± 3.2 IVF-ET-D3: 31.6 ±3.5 IVF-FET-D3: 30.7 ± 4.6; IVF-FET-D5: 29.7 ± 0.6; ICSI-ET-D3: 28 ± 3.6; ICSI-FET-D3: 31.3 ± 4.7; COS: 29.33 ± 1.5	Fetal fraction	Pregnancy	3 (IVF fresh D3), 3 (IVF frozen D3), 3 (IVF frozen D5), 3 (ICSI fresh D3), 3 (ICSI frozen D3), 3 (COS)	/	*H19*, *IGF2*, *SNRPN*	Methylation-specific PCR and pyrosequencing
Jiang et al., 2022 [41]	Cross-sectional	NR	NR/ART: 32.7 ± 3.35; SC: 33.8 ± 3.05	Cord blood	Birth	21	22 (NS)	*MEG3*	Pyrosequencing
Katari et al., 2009 [42]	Cross-sectional	F = 4; M = 2; M + F = 1; Unexp: 3	ART 38.3 ± 5.85; SC: 33.4 ± 7.6/ART: 33.5 ± 7.6; SC: 32.5 ± 4.5	Cord blood Placenta	Birth	10	13 (fertile)	Global DNA methylation	Golden Gate Array
Li et al., 2011 [43]	Cross-sectional	NR	NR/ART: 31.7 ± 3.93; SC:28.9 ± 3.75	Cord blood	Birth	29	30 (NS)	*KvDMR1*, *PEG1*, *H19/IGF2*	DNA bisulfite sequencing
Lim et al., 2009 [44]	Cross-sectional	NR	41.8/36.7	Peripheral blood	Children	25 (11 IVF, 13 ICSI)	87 (NS)	*KvDMR1*, *ZAC*, *PEG1*, *SNRPN*, *DLK1*	Methylation-specific PCR, bisulfite sequencing, pyrosequencing
Litzky et al., 2017 [45]	Cross-sectional	NR	NR/31.5 ± 4.81	Placenta	Birth	18 IVF	158 (NS)	Differences in DNA methylation among groups at the level of 108 imprinted genes	Illumina Infinium Human Methylation 450 array
Liu et al., 2021b [46]	Cross-sectional	NR	NR for al sample/ART: 32.3 ± 5.5; SC: 27.7 ± 2.5	Cord blood	Birth	12 (IVF-ET)	12 (NS)	Global DNA methylation	Human Methylation 450k BeadChip array and bisulfite sequencing
Loke et al., 2015 [47]	Cross-sectional	NR	NR/IVF: 36.9 ± 4.9; SC: 32.2 ± 4.9	Buccal smear	Children	34	174 (fertile)	*LINE1*, *AluYa5*, *H19/IGF2*, *H19*	Mass Array EpiTYPER
Lou et al., 2018 [48]	Cross-sectional	NR	NR	Fetal fraction	Pregnancy	42 COS, 36 IVF, 20 ICSI	/	*H19*, *IGF2*, *SNRPN*	Methylation-specific PCR and pyrosequencing
Mani et al., 2018 [49]	Cross-sectional	NR	35.0–40.5/33.0–36.7	Placenta	Birth	35	35 (NS)	Global DNA methylation	Illumina MethylationEPIC BeadChip array and validation with pyrosequencing
Manning et al., 2000 [50]	Prospective uncontrolled	M	NR	Peripheral blood	Children	92 (ICSI)	/	DNA methylation at 15q11-q13 region (PWS/AS region)	Methyl-specific PCR
Melamed et al., 2015 [51]	Cross-sectional	NR	NR/ART: 38.2 ± 2.8; SC: 36.4 ± 2.3	Cord blood	Birth	10	8 (NS)	Global DNA Methylation	Infinium Illumina Methylation 27 Array; pyrosequencing for HOP gene
Nelissen et al., 2013 [52]	Cross-sectional	M = 28 F = 3 Unexpl = 4	NR	Placenta	Birth	35 (5 IVF, 30 ICSI)	35 (fertile)	*IGF2*, *H19*, *MEG3*, *MEST* α and β, *PEG3*, *SNRPN*, *KCNQ1OT1*	Pyrosequencing
Nelissen et al., 2014 [53]	Cross-sectional	NR	ART: 36.3 ± 5.8; SC: 33.5 ± 5.1/ART: 33.9 ± 4.1; SC: 31.1 ± 4.6	Placenta	Birth	81 (IVF/ICSI + ET)	105 (fertile)	*H19*, *IGF2*, *MEST* α and β, *PHLDA2*, *CDKN1C*	Pyrosequencing
Novakovic et al., 2019 [54]	Cross-sectional	NR	NR	Peripheral blood	Children/Adults	149 infants + 158 adults	58 infants + 75 adults (NS)	Global DNA methylation	Infinium Illumina Methylation Epic Bead Chip array
Oliver et al., 2012 [55]	Cross-sectional	NR	NR	Peripheral blood	Children	66 (34 IVF, 32 ICSI)	69 (NS)	*H19*, *KCNQ1OT1*, *SNRPN*, *IGF2*, *INSL5*, *ARHGAP24*, *STK19*, *NCRNA00282*, *JPH4*, *SYP*, *BEX1*	MSQ-PCR; Bisulfite Sequencing; MeDIP and promoter array; Sequenom MassARRAY EpyTIPER
Penova-Vaselinovic et al., 2021 [56]	Cross-sectional	M = 32.47% F = 43.29% Unexpl = 18.18%	NR/ART: 33.9 ± 3.9 SC: 28.5 ± 5.8	Peripheral blood	Adults	231	1188 (NS)	Global DNA methylation	In the ART group evaluated by and in the SC group by Illumina INfinium Human Methylation BeadChip Array
Pliushch et al., 2015 [57]	Cross-sectional	NR	IVM +ART: 36 ± 4; ART: 36.5 ± 4.5/IVM+ ART: 32.0 ± 1.5; ART: 35.0 ± 4.0	CVS, cord blood	Birth	30 (11 IVM + IVF/ICSI, 19 IVF/ICSI)	/	*LIT1*, *MEST*, *MEG3*, *NESPas*, *PEG3*, *SNRPN*, *APC*, *ATM*, *BRCA1*, *RAD51C*, *TP53*, *NANOG*, *OCT4*, *LEP*, *NR3C1*, *LINE1*, *ALU*	Bisulfite pyrosequencing
Puumala et al., 2012 [58]	Cross-sectional	M = 17.28% F = 21.34% M and F = 16.26% Unexpl. = 6.10%	NR/ART: 34.1 ± 3.9; SC: 29.6 ± 4.3	Buccal smear	Children	67 (IVF/ICSI)	31 (fertile)	*IGF2*, *H19*, *IGF2R*, *KvDMR*	Pyrosequencing
Rancourt et al., 2012 [59]	Cross-sectional	NR	NR/IVF: 36.5 ± 4.5; OI: 34.5 ± 4.6; SC: 35.5 ± 4.7	Placenta, cord blood	Children	86 (27 OI, 59 IVF)	61 (NS)	*MEST*, *GRB10*, *KCNQ1*, *SNRPN*, *H19*, *IGF2*	Pyrosequencing
Rossignol et al., 2006 [60]	Cross-sectional	NR	NR	Peripheral blood	Children	11	29 (NS)	*H19*, *IGF2*, *SNRPN*, *PEG1/MEST*	Southern blot Bisulfite sequencing
Sakian et al., 2015 [61]	Cross-sectional	NR	NR/IVF: 35.3 ± 3.9; ICSI: 34.1 ± 2.9; SC: 32.4 ± 8.7	Placenta	Birth	97 (56 IVF, 41 ICSI)	22 (fertile)	*H19*	Pyrosequencing
Santos et al., 2010 [62]	Cross-sectional	NR	NR	Embryo, blasts	/	138 (75 IVF, 63 ICSI), 27 (14 IVF, 13 ICSI)	/	Global DNA methylation	Anti-5-methyl cytosine antibodies
Shi et al., 2014 [63]	Observational uncontrolled	M = 3/23 F = 20/23	NR	Embryo	/	254	/	*H19*, *PEG1*, *KvDMR*	Bisulfite PCR and pyrosequencing
Song et al., 2015 [64]	Cross-sectional	NR	ART: 36.2 ± 5.3; SC: 34.9 ± 5.7/ART: 35.3 ± 3.7; SC: 34.5 ± 5.0	Placenta	Birth	88	49 (fertile)	DNA methylation of 37 CpG in 16 different genes (*CCDC62*, *CRTAM*, *FLJ10260*, *FLJ90650*, *GRB10*, *GRIN2C*, *H19*, *IL5*, *LYST*, *MEST*, *NDN*, *PCDHGB7*, *PTPN20B*, *SNRPN*, *TCF2*, *TTR*)	Bisulfite DNA and pyrosequencing
Tang et al., 2017 [65]	Cross-sectional	M	NR	Cord blood	Birth	13 ICSI	30 (fertile)	*H19*, *SNRPN*, *KCQ1OT1*	Pyrosequencing
Tierling et al., 2010 [66]	Cross-sectional	NR	NR/IVF: 34.8 ± 4; ICSI: 35.3 ± 4.3; SC: 31.7 ± 5.7	Peripheral blood	Children	112 (35 IVF, 77 ICSI)	73 (NS)	*KvDMR1*, *H19*, *SNRPN*, *MEST*, *GRB10*, *DLK1/MEG*	Bisulfite techniques (SNuPE assay with SIRPH, Homoduplex separation, pyrosequencing)
Turan et al., 2010 [67]	Cross-sectional	NR	NR/ART: 36 ± 4; SC: 31 ± 6	Placenta, cord blood	Children	45	56 (fertile)	*IGF2/H19*	Pyrosequencing
Vincent et al., 2016 [68]	Cross-sectional	NR	NR/NR in total sample	CVS, cord blood	Birth	150 (68 ICSI, 82 IVF)	66 (NS)	*PLAGL1*, *KvDMR1*, *PEG10*, *LINE1*	Bisulfite assay and pyrosequencing
White et al., 2015 [69]	Cross-sectional	NR	NR	Embryo, blasts	/	24 + 29	/	*SNRPN*, *KCNQ1OT1*, *H19*	Bisulfite clonal sequencing
Whitelaw et al., 2014 [70]	Retrospective cohort	NR	NR/ART: 34.6 ± 3.3; SC: 34.1 ± 3.4	Buccal smear	Children	69 (49 IVF-ET, 20 ICSI-ET)	89 (fertile)	*LINE1*, *SNRPN*, *PEG3*, *INS*, *IGF2*	Pyrosequencing
Wong et al., 2010 [71]	Cross-sectional	NR	NR/ART: 36.4 ± 3.1; ICSI: 35.0 ± 4.8; SC: 33.0 ± 4.9	Placenta, cord blood	Children	77 (32 IVF, 45 ICSI)	12 (NS)	*H19*	MS-SNuPE
Yoshida et al., 2013 [72]	Cross-sectional	NR	NR	Placenta, cord blood	Children	8 IVM + IVF	/	*H19*, *GTL2*, *Zdbf2*, *PEG1*, *PEG3*, *LIT1*, *ZAC*, *SNRPN*	Imprinted methylation Assay
Zhang et al., 2019 [73]	Cross-sectional	NR	NR	Cord blood	Birth	33	43 (NS)	*AGTR1*	Bisulfite sequencing

**Abbreviations**. ART, assisted reproductive technique; COS, controlled ovarian stimulation; CVS, chorionic villus sampling; ET, embryo transfer; FET, frozen embryo transfer; ICSI, intracytoplasmic sperm injection; IUI, intrauterine insemination; IVF, in vitro fertilization; OI, ovulation induction; SC, spontaneous conception, NR, not reported. **Genes:***APC*, *Adenomatous Polyposis Coli; AGTR1*, *angiotensin II receptor type 1; ALU*, *Arthrobacter luteus; ARHGAP24*, *Rho GTPase Activating Protein 24; ATM*, *Ataxia-Telangiectasia Mutated; BEX1*, *Brain Expressed X-Linked 1; BRCA1*, *BReast CAncer gene 1; CCDC62*, *Coiled-Coil Domain Containing 62; CDKN1C*, *Cyclin-dependent kinase inhibitor 1C; CRTAM*, *Cytotoxic And Regulatory T Cell Molecule; DLK1*, *Delta Like Non-Canonical Notch Ligand 1; ERVFRD1*, *Endogenous Retrovirus Group FRD Member 1; ERW1*, *Endogenous Retrovirus Group W Member 1; FLJ10260*, *Schlafen Family Member gene; FLJ90650*, *Laeverin gene; GRB10*, *Growth Factor Receptor Bound Protein 10; GRIN2C*, *Glutamate Ionotropic Receptor NMDA Type Subunit 2C; GTL2*, *gene trap locus2; HERV-FRD*, *Human Endogenous Retrovirus FRD; IGF2*, *insuline-like growth factor 2; IL5*, *Interleukin 5; INSL5*, *insulin like 5; JPH4*, *Junctophilin 4; KCNQ1*, *Potassium Voltage-Gated Channel Subfamily Q Member 1; KCNQ1OT1*, *KCNQ1 Opposite Strand/Antisense Transcript 1; KvDMR1*, *Potassium Voltage Differentially Methylated Region 1; L3MBTL*, *Lethal(3) Malignant Brain Tumor-Like protein; LEP*, *Leptin gene; LINE1*, *Long Interspersed Nuclear Elements 1; LIT1*, *Long QT Intronic Transcript 1; LYST*, *Lysosomal Trafficking Regulator; MEG3*, *Maternally Expressed Gene 3; MEST*, *Mesoderm Specific Transcript; NANOG*, *Homeobox protein Nanog; NDN*, *Necdin; NESPas*, *GNAS antisense; NCRNA00282*, *Non-Coding Ribonucleic Acid 00282; NR3C1*, *Nuclear Receptor Subfamily 3 Group C Member 1; OCT4*, *octamer-binding transcription factor 4; PCDHGB7*, *Protocadherin Gamma Subfamily B 7; PEG1*, *Paternally expressed gene 1; PEG3*, *Paternally expressed gene 3; PEG10*, *Paternally expressed gene 10; PHLDA2*, *Pleckstrin Homology Like Domain Family A Member 2; PLA2GA2*, *phospholipase A2 group IIA; PTPN20B*, *protein tyrosine phosphatase non-receptor type 20B; RAD51C*, *Rad recombinase 51 paralog C; SNRPN*, *Small Nuclear Ribonucleoprotein Polypeptide N; SNURF*, *SNRPN Upstream Open Reading Frame; SLC2A3*, *Solute Carrier Family 2 Member 3; STK19*, *Serine/threonine-protein kinase 19; SYP*, *Synaptophysin; TCF2*, *Transcription factor 2 gene; TP53*, *Tumor Protein 53; TTR*, *Transthyretin; VEGFA*, *Vascular endothelial growth factor A; ZAC*, *Zinc-Activated ion Channel; ZDBF2*, *Zinc Finger DBF-Type Containing 2.* NS, non-specified.

**Table 3 jcm-11-05056-t003:** Evaluation of study quality using “The Cambridge Quality Checklists”.

Author and Year of Publication	Checklist for Correlates	Checklist for Risk Factors	Checklist for Causal Risk Factors	Total
Argyraki et al., 2021 [24]	2	1	2	5/15
Barberet et al., 2021 [25]	3	1	2	6/15
Barberet et al., 2021 [26]	2	1	2	5/15
Camprubì et al., 2013 [27]	3	1	2	6/15
Caramaschi et al., 2011 [28]	3	1	2	6/15
Castillo-Fernandez et al., 2017 [29]	2	1	2	5/15
Chen et al., 2018 [30]	2	1	2	5/15
Chen et al., 2020 [31]	3	1	2	6/15
Choufani et al., 2018 [32]	3	1	5	9/15
Choux et al., 2018 [33]	2	1	2	5/15
DeBaun et al., 2003 [34]	2	1	1	4/15
El Hajj et al., 2017 [35]	2	1	2	5/15
Estill et al., 2016 [36]	3	1	2	6/15
Feng et al., 2011 [37]	2	1	2	5/15
Ghosh et al., 2017 [38]	2	1	2	5/15
Gomes et al., 2009 [39]	1	1	2	4/15
Ji et al., 2018 [40]	2	1	1	4/15
Jiang et al., 2022 [41]	2	1	2	5/15
Katari et al., 2009 [42]	2	1	2	5/15
Li et al., 2011 [43]	2	1	2	5/15
Lim et al., 2009 [44]	2	1	2	5/15
Litzky et al., 2017 [45]	2	1	5	8/15
Liu et al., 2021b [46]	2	1	2	5/15
Loke et al., 2015 [47]	1	1	2	4/15
Lou et al., 2018 [48]	3	1	1	5/10
Mani et al., 2018 [49]	3	1	5	9/15
Manning et al., 2000 [50]	2	3	1	6/15
Melamed et al., 2015 [51]	3	1	2	6/15
Nelissen et al., 2013 [52]	2	1	2	5/15
Nelissen et al., 2014 [53]	3	1	2	6/15
Novakovic et al., 2019 [54]	3	1	2	6/15
Oliver et al., 2012 [55]	3	1	2	6/15
Penova-Vaselinovic et al., 2021 [56]	3	1	2	6/15
Pliushch et al., 2015 [57]	3	1	1	5/15
Puumala et al., 2012 [58]	2	1	2	5/15
Rancourt et al., 2012 [59]	2	1	2	5/15
Rossignol et al., 2006 [60]	3	1	2	6/15
Sakian et al., 2015 [61]	2	1	2	6/15
Santos et al., 2010 [62]	2	1	1	4/15
Shi et al., 2014 [63]	1	1	1	3/15
Song et al., 2015 [64]	1	1	2	4/15
Tang et al., 2017 [65]	2	1	2	5/15
Tierling et al., 2010 [66]	3	1	2	6/15
Turan et al., 2010 [67]	2	1	2	5/15
Vincent et al., 2016 [68]	2	1	2	5/15
White et al., 2015 [69]	2	1	1	4/15
Whitelaw et al., 2014 [70]	2	2	5	9/15
Wong et al., 2010 [71]	1	1	2	4/15
Yoshida et al., 2013 [72]	1	1	1	3/15
Zhang et al., 2019 [73]	2	1	2	5/15

## Data Availability

Not applicable.

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
