# Peer review of "DNA Methylation in Offspring Conceived after Assisted Reproductive Techniques: A Systematic Review and Meta-Analysis"

_jcm, 2022, doi:10.3390/jcm11175056_

Round 1

Reviewer 1 Report

The paper is well written and reads well.

However, following are my concerns-

 The authors have put all the required information of the included studies in table 2. The details are described comprehensively and the authors have listed the studies alphabetically (author name).

1. However, the articles can be grouped together based on the type of tissue used. This will help in having better clarity towards understanding the DNA methylation changes in the placenta, then cord blood, then in peripheral blood of children mentioning the age group and then followed by studies in adults. .

Following this, the articles in each tissue group can be sub categorized based on the type of ART protocol used.

2. Another major concern is regarding the control or the spontaneously conceived group.

It is important to understand whether subjects under this group were clinically infertile but conceived spontaneously, indicating that these studies are confounded by the clinical infertility status parents in control group.

The authors therefore need to add one more column in table 2 indicating the details on this.

3. Line 144, put full stop after ‘controlled samples’

Author Response

Manuscript ID: jcm-1846663

General comments

The paper is well written and reads well.

However, following are my concerns. The authors have put all the required information of the included studies in table 2. The details are described comprehensively and the authors have listed the studies alphabetically (author name).

Comment 1: However, the articles can be grouped together based on the type of tissue used. This will help in having better clarity towards understanding the DNA methylation changes in the placenta, then cord blood, then in peripheral blood of children mentioning the age group and then followed by studies in adults. .

Following this, the articles in each tissue group can be sub categorized based on the type of ART protocol used.

Answer to comment 1: We appreciated your comment. Studies are grouped based on the tissue in the plots. Indeed, each outcome was sub-analyzed, based on the tissue. Regarding Table 2, we would prefer to keep it as is since we have not reported the results of the studies in this table. Therefore, the reorganizing of information would not lead to a better understanding of the changes in DNA methylation.

Comment 2: Another major concern is regarding the control or the spontaneously conceived group.

It is important to understand whether subjects under this group were clinically infertile but conceived spontaneously, indicating that these studies are confounded by the clinical infertility status parents in control group. The authors therefore need to add one more column in table 2 indicating the details on this.

Answer to comment 2: Thank you for reporting this important issue. We carefully reviewed all included studies, shown in table 2. We avoided including an additional column. We used the “SC group” column to report the parent’s fertility status (in parentheses). When the study reported this information, we wrote: “fertile” or “infertile” (in fact, none of the studies included infertile couples as controls). When the information was not given, we reported “NS” (non-specified). You will find any changes highlighted in yellow. We hope this satisfactorily addresses your concern.

Comment 3: Line 144, put full stop after ‘controlled samples’

Answer to comment 3: Added, thank you.

Reviewer 2 Report

This systematic review presents the available information found in their literature search regarding the effects of ART procedures on the methylation of global DNA and specific imprinted genes. The methodology of their systematic review is clearly presented and the data is logically analyzed presenting the current knowledge while the problematic aspects of the subject matter is emphasized. Both global methylation and methylation of imprinted genes may be susceptible to a great variety of factors, intrinsic and extrinsic, contributing to the confounding results reported in the literature, mainly due to the complicated feasibility of focused research. As the matter of possible side effect of ART on the health of the offsprings  due to effects on their epigenetic status is important, the presented manuscript adds valuable information to the knowledge base, indication the need for further research.

Author Response

Manuscript ID: jcm-1846663

General comments

This systematic review presents the available information found in their literature search regarding the effects of ART procedures on the methylation of global DNA and specific imprinted genes. The methodology of their systematic review is clearly presented and the data is logically analyzed presenting the current knowledge while the problematic aspects of the subject matter is emphasized. Both global methylation and methylation of imprinted genes may be susceptible to a great variety of factors, intrinsic and extrinsic, contributing to the confounding results reported in the literature, mainly due to the complicated feasibility of focused research. As the matter of possible side effect of ART on the health of the offsprings due to effects on their epigenetic status is important, the presented manuscript adds valuable information to the knowledge base, indication the need for further research.

Answer to General comments: We greatly appreciated your comment and the time you took to review our manuscript.